# Availability, affordability and price components of insulin products in different-level hospital pharmacies: Evidence from two cross-sectional surveys in Nanjing, China

Lulu Wang[1]*, Liang Dai[2], Hui Liu[1], Huizhen Dai[3], Xin Li[4], Weihong Ge[1]

1 Department of Pharmacy, Nanjing Drum Tower Hospital, The Affiliated Hospital of Nanjing University Medical School, Nanjing, Jiangsu, China, 2 Laboratory Management Office, China Pharmaceutical University, Nanjing, Jiangsu, China, 3 Jiangsu Medicine Information Institute, Nanjing, Jiangsu, China, 4 Department of Clinical Pharmacy, School of Pharmacy, Nanjing Medical University, Nanjing, Jiangsu, China

* hersheylulu@163.com

**Data Availability Statement:** All relevant data are within the manuscript and its Supporting Information files.

## Abstract

The essential medicine——insulin cannot be easily accessed and afforded in many countries. To help address this issue, we evaluated the availability, affordability and price of insulin products in Nanjing, eastern China. Two cross-sectional studies were conducted in 2016 and 2018. A total of 56 hospital pharmacies were sampled, using a simplified and adapted World Health Organization/Health Action International (WHO/HAI) methodology. Prices were expressed as Median Price Ratios (MPRs) to Australian Pharmaceutical Benefit Scheme (PBS) prices. In addition, we investigated the price components of seven selected insulin products as a case study before and after the Online Centralized Procurement Policy for Hospital Drugs in May, 2018. Affordability was presented as the number of daily wages of the lowest paid unskilled government worker (LPGW) required to purchase 1000IU of insulin based on the average courses of treatment, approximately 30 days' treatment. The availability of insulin products was very high in secondary hospitals and tertiary hospitals both in 2016 and 2018, but in community hospitals was very low. In 2018, the availability of prandial insulin products showed fluctuation compared to 2016. The availability of pre-mixed human insulin products was over 95% overall, and also very high (80%) in community hospitals in 2018. The prices of insulin products were much lower than PBS prices of Australian in this study, with the MPRs less than 1 (0.32 to 0.71 in 2016 vs. 0.30 to 0.68 in 2018) for all insulin types. But insulin products in Nanjing in 2016 and 2018 were considered unaffordable, because the number of daily wages of the LPGW needed to purchase for the 30 days treatment of insulin products ranged from 2.26 to 8.49 in 2016 and 1.88 to 7.09 in 2018. The manufacturers' selling price contributed the main part (74.15% to 77.70% before and 74.86% to 91.51% after the implementation of the bidding policy) of the price components of target insulin brands. The availability of insulin products was high in secondary hospitals and tertiary hospitals, but lower in community hospitals. However, the affordability in community hospitals was better than other hospitals, but the insulin products were still unaffordable for patients on low incomes. Further improvements of the availability accessibility and

**Funding:** The authors received no specific funding for this work.

**Competing interests:** The authors have declared that no competing interests exist.

affordability of medicines in advancing health insurance policies and lowering drug prices should be put forward.

## Background

Globally, approximately 80% deaths were attributable to chronic diseases each year, among which 1.5 million deaths (2% of all deaths) were caused by diabetes [1, 2]. Before the discovery of insulin, diabetes was considered as an incurable disease, and killed tens of millions of people every year, leading to serious complications [3]. Studies revealed that the morbidity and mortality of chronic disease would be effectively controlled when medications were made accessible and affordable [4, 5]. Similarly, patients with diabetes who received effective treatment may lead to substantial reductions in morbidity and mortality [6–8].

Currently, about one in two people who need insulin lack access to the life-saving medicine [9]. This is caused by many reasons, such as low availability, poor affordability, prescription irregularities and poor patient adherence, etc. [3]. The cost of insulin accounts for 70% of the total cost of diabetes treatment [10]. But the routine availability of insulin is not optimistic in many parts of the world, even in many developed countries because of its high cost [11, 12]. Patients with diabetes usually have poor health, low quality of life and more work delays, which may lead to unemployment. They need to use more medical resources, and the more the complications are, the more serious the situation is [13].

According to the report, there are 113.9 million people suffering from diabetes and 493.4 million people with prediabetes among Chinese adults aged 18 and above [14]. The prevalence rate of diabetes in Chinese adults was 11.6% and 50.1% was prediabetes patients [14]. At present there were steadily increasing diabetes burden in China, coupled with poor management and increasing complications such as cardiovascular diseases [15].

In order to understand the availability and affordability of insulin products, and for diabetics to use insulin more conveniently. We selected Nanjing, a representative city in the eastern China, as the data acquisition site to conduct our study. Although there have been studies assessed the access to insulin in central China [16] and western China [17], but none in eastern China where the economic and drug management level are above the average of China. This study is about two cross-sectional surveys of the price, availability and affordability of insulin products from different-level hospital pharmacies in Nanjing, Jiangsu province first in 2016 and again in 2018. And the price components of insulin products also investigated along the supply chain.

## Method

### Study setting

Nanjing is the capital city of Jiangsu province, a developed region in eastern China, with a population of 8.43 million and a huge burden of diabetes mellitus [18]. Diabetes (death ratio = 3.41%) is the 6th leading cause of death in Nanjing city [19]. The annual mortality rate of diabetes (per 100 000 people) in Nanjing city is 20.96, which has increase to 43.07% since 1999 [20].

Insulin is prescription drug in China which requires a prescription from an authorized doctor. Hospitals are the main places where patients can obtain insulin products, because the most practicing doctors are employed by the institutions and patients are encouraged to have prescription dispensed in these outlets. Patients can get their medicines directly from the

hospital pharmacy without a referral. Therefore, the price, availability and affordability of insulin products in different-level hospital pharmacies are worth discussing.

Despite the implementation of the national health insurance and its universal coverage (more than 95% [21]), a large proportion for outpatient drugs is still required to pay for. To reduce the burden on patients, China began implementing a National Essential Medicine System in 2009, which aims to guarantee the use of drugs by patients. Over the years, China's National Essential Medicine System has played an important role in the medical security reform, medicine production and sales, and clinical specification of use [22]. In addition, the 'zero-mark-up' policy has applied to primary care facilities since 2012 [23]. Nanjing has implemented the 'zero-mark-up' policy for medicines in all public hospitals since October, 31, 2015 [24], which canceled the 15% mark-up of the hospital pharmacy, lowered drug prices, reduced the burden of patients effectively.

## Design

Two cross-sectional studies of insulin products availability and affordability in different-level hospital pharmacies were conducted to make comparisons between 2016 and 2018, using a simplified and adapted WHO/ Health Action International (HAI) method [25]. In addition, we investigated the price components of seven selected insulin products as a comparison study before and after the implementation of the Online Centralized Procurement Policy for Hospital Drugs in May, 11, 2018 [26–28].

As the capital city of Jiangsu province, Nanjing can be regarded as a high-income city. According to the Classification Management Standards for General Hospitals [29], hospitals are divided into three levels, community hospitals, secondary hospitals and tertiary hospitals respectively, in line with their tasks and functions. Community hospitals are primary medical and health institutions that provide medical treatment, prevention, health care and rehabilitation services directly to communities with a certain population ($\leq$ 100 000). Secondary hospitals are health institutions that provide services to multiple communities (the radius population is generally more than 100 000). They are the middle level of the three-level medical and health system. Tertiary hospitals are regional or above hospitals that provide high-level specialized medical and health services and carry out higher education and scientific research tasks in several regions.

Nanjing city is divided into 13 administrative regions, include 11 districts and 2 counties. There are 20 tertiary hospitals, 11 secondary hospitals and 25 community hospitals in all of the regions (Table 1), all of the 56 hospitals were included in this survey.

## Data collection

The sales data of all medicines in Nanjing were collected by Jiangsu Medicine Information Institute annually. Two data analysts after training (L. Wang and L. Dai) screened and separated the sales data of insulin products from different-level hospital pharmacies in Nanjing in 2016 and 2018 by using an efficiently designed data processing form. The items in the designed form as insulin type, brand name, strength [International Unit per milliliter (IU/ml)], dosage form (presentation of insulin products, such as vial, cartridge or prefilled pen), volume (ml), pack size, manufacturer and price for patients were collected. All the data were checked by the supervisor (X. Li) for accomplishment.

**Table 1. Main characteristics of Nanjing's survey districts and number of sample facilities.**

| Number of regions | Population (million) | GDP of 2018 ($) | Community hospitals | Secondary hospitals | Tertiary hospitals |
|---|---|---|---|---|---|
| 11 | 8.43 | 186 billion | 25 | 11 | 20 |

The insulin products were grouped into three sub-types: prandial (short-acting and rapid-acting insulin), basal (intermediate-acting and long-acting insulin), and pre-mixed insulin according to the onset and duration of action. The data included human and analogue insulin products, but animal insulin products were rarely used in hospitals in Nanjing, so they were not involved in this study.

Then we conducted with the pharmacists of the pharmaceutical purchasing department by face-to-face interviews in order to investigate the price components (mark-ups or taxes) of seven selected insulin products before and after May, 11, 2018 as a comparison study. The prices of manufacturer and patient were recorded in the supply chain, and were tracked back from the selling price of pharmacies in hospitals to the selling prices of manufacturers. These seven insulin products include two human insulin products and five analogues, six of which were imported and one was locally produced (Table 2).

## Data analysis

**Availability.**   The WHO/HAI method was adapted to investigating the data in this study. Availability of the selected medicines were defined as the percentage of pharmacies where insulin products could be accessed in the year of the survey. The availability of insulin products was compared in different sub-types using descriptive statistics. Availability was classified as follow ranges as reported [30]:

<30% very low;

30%—49% low;

50%—80% fairly high;

>80% very high.

**Price.**   We used Median Price Ratio (MPR) to express prices of insulin products according to WHO/HAI method, which is a ratio of the local median price of selected medicines during the survey to the international reference prices (IRPs) (see the formula below). The *International Medical Products Price Guide* from Management Sciences for Health (MSH) is recommended as the most commonly used reference price by the WHO/HAI methodology [31]. However, the prices of analogue insulin products were not included in the *Guide* so was of limited value. We adopted the Australia Pharmaceutical Benefit Scheme (PBS) as international reference prices by resorted to other sources in the literatures [16, 17]. Because the PBS prices represent reimbursement prices paid by the Australian government so it could be regarded as the patient price [32], while MSH prices represent the procurement price of the buyer. Both local prices and the IRPs were converted to US dollars ($) using the exchange rate of the first day of data collection according to OANDA (https://www.oanda.com/currency/converter).

$$MPR = \frac{price\ of\ 100\ IU\ insulin\ in\ Nanjing\ in\ US\$}{price\ of\ 100\ IU\ insulin\ Australia\ PBS\ in\ US\$}$$

**Table 2. The insulin products for the survey of price component.**

| Brand name | Type | Strength | Dosage form | Manufacturers |
|---|---|---|---|---|
| Novolin R | Human | 3ml:300IU | Cartridge | Novo Nordisk |
| Humalog | Analogue | 3ml:300IU | Cartridge | Lilly |
| NovoRapid | Analogue | 3ml:300IU | Cartridge | Novo Nordisk |
| Lantus | Analogue | 3ml:300IU | Pre-filled pen | Sanofi-Aventis |
| Levemir | Analogue | 3ml:300IU | Pre-filled pen | Novo Nordisk |
| Novolin Mix 30 | Human | 3ml:300IU | Cartridge | Novo Nordisk |
| Changxiulin | Analogue | 3ml:300IU | Cartridge | Gan Lee Pharmaceutical co. Ltd |

Generally, the evaluating criterion of MPR revealed that an MPR of 1 or less represents efficient procurement in the public hospitals [33]. We used it as a reference when assessing patient price, whereas this criterion is suitable for evaluating procurement price.

**Affordability.** As the approach HAI recommended [25], affordability of insulin products was expressed as the number of daily wages of the lowest paid unskilled government worker (LPGW) required to purchase 1000 IU of insulin based on the average courses of treatment [34].

In 2018, the average salary of unskilled government workers in Nanjing is US$235.52 per month, with an average daily wage of US$7.85 [35]. According to the National Essential Drugs Prescription Book, the average treatment courses of adult chronic diseases is approximately 30 days. The index evaluation criterion of affordability refers to the ratio between the cost of the treatment and the minimum daily wage standard in the region is less than 1, then the medicine is affordable, otherwise, it is not.

We compared differences in availability and affordability of insulin products surveyed between 2016 and 2018. Moreover, different dosage forms, different types of insulin products, different suppliers were compared between the two years. Microsoft Excel was used to enter data. The normality and homogeneity of variances of prices and affordability were tested, and the differences of those indicators across the pharmacies were compared by t-test for parametric analyses or Kruskal-Wallis test for nonparametric analyses. SPSS 20.0 was used for statistical analyses and a $p$ value $< 0.05$ was considered as significant.

**Price components.** Price components of insulin products were recorded as following stages: manufacturer selling price (MSP), importer selling price, wholesaler selling price, outlet selling price or patients purchase price, and taxes. The changes in price components of insulin products were compared before and after the implementation of the Online Centralized Procurement Policy for Hospital Drugs.

## Results

In the 56 hospital pharmacies sampled, a total of 30 insulin products were found during the survey. 529 price points were recorded in 2016, and 550 in 2018. As shown in Table 3, the dosage form is mainly based on cartridges containing 300 IU/3ml (80.72% in 2016 vs. 77.82% in 2018), both the prefilled pen (13.42% in 2016 vs. 18.00% in 2018) and vials have their parts (5.86% in 2016 vs. 4.18% in 2018). The types of insulin products found in the pharmacies is presented in Table 4, human insulin and analogues were included, each take up about half (human insulin with 54.05% in 2016 and 42.36% in 2018; analogues with 45.95% in 2016 and 57.64% in 2018). Most of the insulin products were imported, 94.70% in 2016 and 92.73% in 2018 (as shown in Table 5).

### Availability

The availability of insulin products was generally higher in secondary hospitals and tertiary hospitals than community hospitals, especially in the basal insulin products each year (Tables 6 and 7).

In 2016, the three sub-types of insulin products could be well found in over 75% of the hospitals sampled, except basal insulin in community hospitals (human short-acting 12%, human

**Table 3. Dosage form of insulin products found in the pharmacies.**

| Characteristics | | Number of products in years | |
|---|---|---|---|
| | | **2016** | **2018** |
| **Presentation** | Injection, containing 400IU/10ml | 5.86% | 4.18% |
| | Cartridge, containing 300IU/3ml | 80.72% | 77.82% |
| | Prefilled Pen, containing 300IU/3ml | 13.42% | 18.00% |

**Table 4. Types of insulin products found in the pharmacies.**

| Characteristics | | Number of products in years | |
|---|---|---|---|
| | | **2016** | **2018** |
| **Type of insulin** | **Human insulin** | | |
| | Short-acting | 15.31% | 6.91% |
| | Intermediate-acting | 13.98% | 5.45% |
| | Pre-mixed | 24.76% | 30.00% |
| | **Analogue insulin** | | |
| | Rapid-acting(Aspart) | 9.83% | 10.00% |
| | Rapid-acting(Lispro) | 4.16% | 5.64% |
| | Long-acting(Glargine) | 10.78% | 11.82% |
| | Long-acting(Determir) | 4.54% | 5.82% |
| | Pre-mixed(Aspart/protamine) | 9.83% | 12.91% |
| | Pre-mixed(Lispro/protamine) | 6.81% | 11.45% |

intermediate-acting 1%, long-acting analogue 44% and pre-mixed analogue insulin 48%). Although the availability of human short-acting insulin (12%) and human intermediate-acting insulin (1%) were very low and both long-acting analogue (44%) and pre-mixed analogue insulin (48%) products were low in community hospitals, they were fairly highly available (more than 70%) in secondary hospitals and tertiary hospitals. And about 70% of hospitals had rapid-acting analogues and pre-mixed human insulin products.

As the data in 2018, the availability of prandial insulin products showed fluctuation compared to 2016, with the reduction of human short-acting insulin and increase of rapid-acting analogue. While the availability of human short-acting insulin (8%) was still very low in community hospitals in 2018.This situation also appeared in basal insulin products. The availability of human intermediate-acting insulin increased slowly (4% in 2016 to12% in 2018) in community hospitals, meanwhile the availability of long-acting analogues grew (44% in 2016 to 60% in 2018). Insulin products also remained highly available in secondary hospitals and tertiary hospitals. As we can see, pre-mixed insulin was highly likely to be prescribed in all hospitals in recent years. The availability of pre-mixed human insulin products were over 95% overall, and also very high (80%) in community hospitals in 2018.

## Price and affordability

The price survey in Nanjing revealed that prices of insulin products were much lower than PBS prices of Australian, with the MPRs less than 1 (0.32 to 0.71 in 2016 vs. 0.30 to 0.68 in 2018) for all insulin types. The number of daily wages of the lowest paid unskilled government

**Table 5. Suppliers of insulin products found in the pharmacies.**

| Characteristics | | Number of products in years | |
|---|---|---|---|
| | | **2016** | **2018** |
| **Supplier** | Imported | 94.7% | 92.73% |
| | Sanofi-Aventis | 7.37% | 8.00% |
| | Bayer | 2.27% | 1.09% |
| | Lilly | 28.73% | 28.91% |
| | Novo Nordisk | 56.33% | 54.73% |
| | Locally produced | 5.3% | 7.27 |
| | Gan Lee Pharmaceutical co. Ltd | 3.41% | 3.82% |
| | Dongbao Pharmaceutical co. Ltd | 1.89% | 3.45% |

**Table 6. Mean availability (%) of insulin products in different-level of pharmacies.**

| Year 2016 | Type of insulin | | Tertiary hospital | Secondary hospital | Community hospital | p |
|---|---|---|---|---|---|---|
| | Prandial insulin | Overall | 0.9500 | 0.9091 | 0.7600 | 0.190 |
| | | Human short-acting | 0.8500 | 0.9091 | 0.1200 | <0.001 |
| | | Analogue rapid-acting | 0.9000 | 0.8182 | 0.7200 | 0.282 |
| | Basal insulin | Overall | 0.9500 | 0.9091 | 0.0450 | <0.001 |
| | | Human intermediate-acting | 0.7500 | 0.7273 | 0.0400 | <0.001 |
| | | Analogue long-acting | 0.9500 | 0.9091 | 0.4400 | <0.001 |
| | Pre-mixed insulin | Overall | 0.9500 | 0.9091 | 0.8800 | 0.836 |
| | | Human | 0.9000 | 0.9091 | 0.8800 | 1 |
| | | Analogue | 0.9000 | 0.9091 | 0.4800 | 0.002 |
| Year 2018 | Type of insulin | | Tertiary hospital | Secondary hospital | Community hospital | p |
| | Prandial insulin | Overall | 1.0000 | 1.0000 | 0.8400 | 0.131 |
| | | Human short-acting | 0.7500 | 0.8182 | 0.0800 | <0.001 |
| | | Analogue rapid-acting | 1.0000 | 1.0000 | 0.8400 | 0.131 |
| | Basal insulin | Overall | 1.0000 | 1.0000 | 0.6800 | 0.003 |
| | | Human intermediate-acting | 0.6500 | 0.7273 | 0.1200 | <0.001 |
| | | Analogue long-acting | 0.9500 | 1.0000 | 0.600 | 0.003 |
| | Pre-mixed insulin | Overall | 0.9500 | 1.0000 | 1.0000 | 0.554 |
| | | Human | 0.9500 | 1.0000 | 1.0000 | 0.554 |
| | | Analogue | 0.9000 | 1.0000 | 1.0000 | 0.159 |

worker need to purchase for the 30 days treatment of insulin products ranged from 2.26 to 8.49 in 2016 and 1.88 to 7.09 in 2018 depending on the insulin types and hospitals. The result revealed that all types of the insulin products in Nanjing in 2016 and 2018 were considered unaffordable according to HAI method.

As the data shown in Tables 6–8, in each sub-types of insulin, analogue insulin products were higher in prices and less affordable than human insulin products. Long-acting analogues had the highest price, with the MPRs ranged from 0.55 to 0.71, and required 6.32 to 8.49 daily wages for purchasing long-acting analogues across the sectors, compare to 1.88 to 3.39 days for other insulin in this survey.

Although the differences were not statistically significant, insulin products dispensed from community hospitals were lower price and more affordable than those dispensed from secondary hospitals and tertiary hospitals intuitively. Prices of insulin products were almost equal in the hospitals sampled, but long-acting analogues were not included in.

**Table 7. Mean availability (%) of insulin products in different-level of pharmacies.**

| Type of insulin | | Tertiary hospital | | | Secondary hospital | | | Community hospital | | |
|---|---|---|---|---|---|---|---|---|---|---|
| | | 2016 | 2018 | P | 2016 | 2018 | P | 2016 | 2018 | P |
| Prandial insulin | Overall | 19(95.00) | 20(100) | 0.157 | 10(90.91) | 11(100) | 0.637 | 19(76.00) | 21(84.00) | 0.480 |
| | Human short-acting | 17(85.00) | 15(75.00) | | 10(90.91) | 9(81.82) | | 3(12.00) | 2(8.00) | |
| | Analogue rapid-acting | 18(90.00) | 20(100) | | 9(81.82) | 11(100) | | 18(72.00) | 21(84.00) | |
| Basal insulin | Overall | 19(95.00) | 20(100) | 0.157 | 10(90.91) | 11(100) | 0.157 | 9(4.50) | 17(68.00) | <0.001 |
| | Human intermediate-acting | 15(75.00) | 13(65.00) | | 8(72.73) | 8(72.73) | | 1(4.00) | 3(12.00) | |
| | Analogue long-acting | 19(95.00) | 19(95.00) | | 10(90.91) | 11(100) | | 11(44.00) | 15(60.00) | |
| Pre-mixed insulin | Overall | 19(95.00) | 19(95.00) | 0.157 | 10(90.91) | 11(100) | 0.157 | 22(88.00) | 25(100) | 0.024 |
| | Human | 18(90.00) | 19(95.00) | | 10(90.91) | 11(100) | | 22(88.00) | 25(100) | |
| | Analogue | 18(90.00) | 18(90.00) | | 10(90.91) | 11(100) | | 12(48.00) | 20(100) | |

**Table 8. Median Price Ratio (MPR) and average affordability of insulin in different-level of pharmacies.**

| Type of insulin | | Indicators | Tertiary hospital | | Secondary hospital | | Community hospital | |
|---|---|---|---|---|---|---|---|---|
| | | | *2016* | *2018* | *2016* | *2018* | *2016* | *2018* |
| **Prandail insulin** | **Human short-acting** | **MPR** | 0.34 | 0.32 | 0.33 | 0.31 | 0.32 | 0.30 |
| | | **Affordability** | 2.38 | 1.99 | 2.33 | 1.93 | 2.26 | 1.89 |
| | **Analogue rapid-acting** | **MPR** | 0.37 | 0.38 | 0.35 | 0.37 | 0.38 | 0.36 |
| | | **Affordability** | 3.15 | 2.80 | 2.95 | 2.78 | 3.23 | 2.67 |
| **Basal insulin** | **Human intermediate-acting** | **MPR** | 0.34 | 0.32 | 0.33 | 0.33 | 0.33 | 0.32 |
| | | **Affordability** | 2.38 | 1.97 | 2.33 | 2.01 | 2.33 | 1.97 |
| | **Analogue long-acting** | **MPR** | 0.71 | 0.67 | 0.64 | 0.68 | 0.55 | 0.61 |
| | | **Affordability** | 8.49 | 7.00 | 7.63 | 7.09 | 6.59 | 6.32 |
| **Pre-mixed insulin** | **Human** | **MPR** | 0.33 | 0.31 | 0.33 | 0.32 | 0.32 | 0.33 |
| | | **Affordability** | 2.33 | 1.88 | 2.30 | 1.94 | 2.28 | 2.01 |
| | **Analogue** | **MPR** | 0.40 | 0.36 | 0.37 | 0.37 | 0.35 | 0.38 |
| | | **Affordability** | 3.39 | 2.62 | 3.15 | 2.74 | 2.95 | 2.80 |

Between the year 2016 and 2018, there were tiny fluctuations in the price of insulin, but the affordability showed a trend of increasing after the reimbursement of universal health insurance. The affordability of each type of insulin product in 2018 showed improvement for patients in Nanjing, compared with that in 2016.

## Price components

For each of the seven selected insulin products, the price components were categorized into five stages according to HAI method: manufacturer selling price (MSP), import tariff or purchase taxes, importer mark-ups, wholesaler mark-ups and outlet mark-ups. We found that the MSP contributed the main part (74.15% to 77.70% before and 74.86% to 91.51% after the implementation of the bidding policy) of the price components of target insulin brands in each year.

Fig 1 shows the contributions of each price component to the final purchase price of the seven selected insulin products for patients before the implementation of the bidding policy, and Fig 2 shows the price components after.

## Discussion

In this study, we adopted the WHO/HAI method to estimate the availability, price, affordability and price component of insulin products in 20 tertiary hospitals, 11 secondary hospitals and 25 community hospitals in Nanjing city. According to our survey, we found the availability of most insulin products was fairly high in different-level hospital pharmacies in Nanjing city. Insulin products were more accessible in secondary hospitals and tertiary hospitals than in community hospitals. There was a systematic study showed that the mean availability was 55% to 80% for human insulin and 55% to 63% for analogue insulin in 13 low-income and middle-income countries [36]. But this figure still does not meet the WHO target of 80% availability of affordable essential medicines for non-communicable diseases in any sector [37]. In our survey, the pre-mixed insulin products had a good availability but for the basal insulin products was suboptimal. The data of 2016 and 2018 showed a growing trend of availability of insulin in Nanjing.

The price level of insulin products in Nanjing was not very high, but with an unsatisfactory affordability. Especially for long-acting analogues, which cost about a week's wage of the

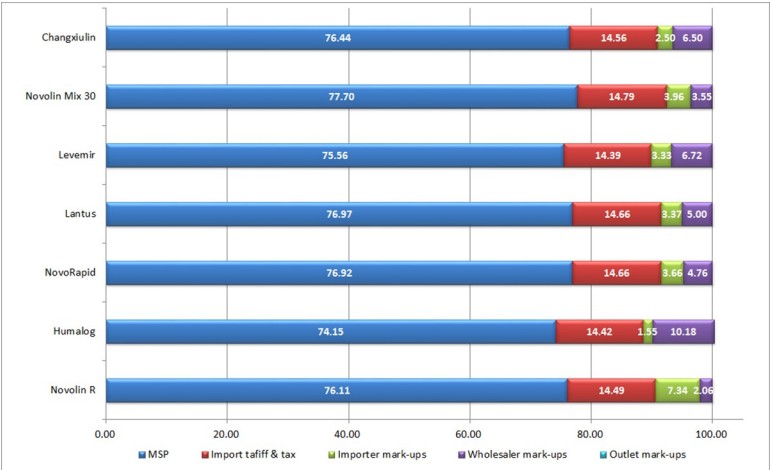

**Fig 1. Insulin price components before the implementation of the bidding policy.**

LPGW to afford 30 days treatment. And the main part of the price components was MSP, which accounts for more than 74% of the final purchase price.

## Availability

Although the community hospital pharmacy is the most convenient and timesaving place to obtain treatment for diabetes patient, the availability of insulin products in these hospitals was not satisfactory. The situation was consistent with the previous government survey in Shaanxi Province [38]. The insulin treatment for type 1 and many type 2 diabetes patients is highly individualized due to the variety of living habits, dietary habits, education level and compliance. A combination use of different types of insulin is recommended, which indicates that the insulin products are widely needed in different-level pharmacies. But only secondary hospitals and tertiary hospitals can meet the need for insulin treatment in recent years from our survey.

Subject to the prescribing limitation of medical insurance in many public hospitals, patients have to attend to the hospitals for consultation and prescription of insulin products frequently.

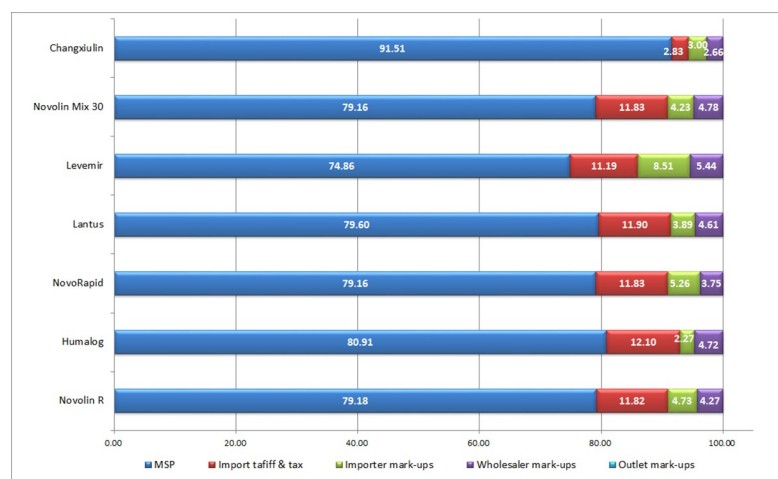

**Fig 2. Insulin price components after the implementation of the bidding policy.**

With overcrowded patients in secondary hospitals and tertiary hospitals and relatively limited medical resources, it is very inconvenient for those who live far away, especially for the elderly and children to access the services, resulting in the waste of unnecessary expenses and time.

## Affordability

Previously, hospitals levied 10–20% mark-ups on the procurement price of medicines to the patients. Since 31 October 2015, the mark-ups of medicines in public hospitals in Nanjing had been cancelled, and patients purchase medicines at their procurement prices [39]. The selling prices of insulin products were much cheaper in public hospitals when our survey conducted.

MPRs in this survey showed an opposite side of the research in Hubei and Shaanxi Province [16, 17]. All types of insulin products were efficient procured in different-level hospitals referenced to the Australian PBS price, but the affordability was sub-optimal in Nanjing when using the number of daily wages of the LPGW had to pay for the 30 days treatment of insulin products the LPGW method. Affordability was ranked from high to low in terms of community hospitals, secondary hospitals and tertiary hospitals. Despite the better affordability, the quality of outpatient services in community hospitals are relatively low, and procurement of some types of insulin is low due to financial and storage constraints, resulting in poor availability.

The survey showed that human insulin products had a better affordability than analogues in three sub-types of insulin, although there is no significant clinical advantages in analogue over human insulin [40].

More than 1 day's wage of the LPGW had to pay for the treatment that lasted 30 days. And in fact, the burden of diabetes patients may be even heavier because patients may have complications at the same time and need more than one type of medication, especially patients on lower income than the LPGW from rural areas. Although there has a universal coverage of health insurance in China, but the benefits of out-patients are limited and the offset for out-patient medication can be insufficient [41].

From the perspective of dosage form, the preparation process of vials is simpler and the price of insulin in vials was lower than cartridges and prefilled pens. But cartridges and pre-filled pens are more convenient in dose adjustment, easier in self-administration, less injection pain and less wastage [42]. These advantages make cartridges and prefilled pens occupy more share of the insulin market not only in Nanjing but all over the world [36], although the WHO advises that insulin in vials should be available in outlets to meet the basic demands.

## Price components

Because of the implementation of the Online Centralized Procurement Policy for Hospital Drugs in May 2018 [26–28], there has been significant adjustments in import tariff or purchase taxes and the mark-ups. The results of our comparative study on price components before and after the implementation of the policy can reflect the effect of the policy to a certain extent and provide references for further policy guidance. Research on the price components of insulin products can provide insight into its price composition and provide evidence-based support for rational drug pricing.

The MSP accounts for the majority of the final price of insulin products (more than 74%), which should be reduced given the large number of diabetic patients in China [14, 15]. Price of medicines in China are centrally negotiated and priced by the government, but the price setting systems in China are currently decentralized and fragmented, jeopardizing the bargaining effect of the government. The relevant government departments lacked an effective method to estimate the production cost of medicines, which can only be priced on the basis of the costs

reported by the manufacturers. And the lack of local manufacturers also made price negotiations difficult. Therefore, we suggest to strictly accounting of drug costs, improving manufacturing price monitoring system, setting the pricing standard and implementing hierarchical and reasonable pricing.

Before the implementation of the policy, import tariff or purchase taxes contributed about 14% of the final price of insulin products, which is not fair to impose it on diabetic patients that mostly children and the elderly. But the purchase taxes of local products reduced to 2.83% and the import tariff of imported products reduced to about 11% after the new policy was implemented. Therefore, the reduction of insulin prices can start with an exemption of insulin products from taxation.

Due to the particularity of the dosage form, the storage and transportation costs of insulin products are higher, so the price mark-ups of the wholesale and retail links is higher than that of ordinary drugs.

Since the mark-up policy in public hospitals was cancelled in 2015, the price of insulin products had been reduced by 15%. The prices of insulin products were cheaper in community hospitals than in secondary hospitals and tertiary hospitals. This may be caused by the competition of price between different-level hospitals. Community hospitals have to lower their prices to attract more patients to prescribe drugs.

## Trends in this survey

From the data of the two years (2016 and 2018), we can see the availability of most insulin products was high in different-level hospital pharmacies and was on the rise. Even the situations of lower ones such as short-acting, intermediate-acting human insulin and both long-acting and pre-mixed analogue products in community hospitals have improved. And in the absence of significant changes in MPR, the affordability of insulin has improved partly because of the continuous efforts of government departments, and partly related to the increase of people's income.

In terms of insulin types, human insulin products still accounted for the main parts. The number of analogue products was increasing, because such advantages as doing a better job in lowering fasting blood sugar, lower risk of hypoglycemia and more convenience of usage. The market of insulin products was still dominated by imports, and local products were growing slowly. The decrease of vials usage due to the disadvantages discussed earlier.

## Limitation

This survey also has several limitations. *Firstly*, The Jiangsu Province is located in the east of China, which is in the middle-upper level of development of China. We only chose Nanjing city as the research object and hence may not be representative of the whole of Jiangsu province. But Nanjing is a typical city which can represent the economically developed areas of Eastern China to some extent. *Secondly*, in this survey, we choose the PBS prices from Australia as international reference prices rather than IPR recommended by WHO/HAI, which may lead to inaccuracy in prices estimated. *Thirdly*, MSP was back estimated from the procurement price of the wholesale and Chinese common tariff because of unable to contact with insulin manufacturers overseas. *Fourthly*, insulin products were more affordable for the patients in the high-income region than the low-income region. For patients in rural region, the affordability of insulin products for a one-month treatment was worse than for urban patients. There may be a certain number of people earn less than LPGW in some rural region hence the affordability may be underestimated. *Finally*, other costs have not be involved in the affordability estimation, such as needing to take more than one medicine, the examinations, syringe,

storage of insulin products, glucometers and test trips, travelling expenses, accommodation, catering, transportation costs, loss of productivity and income, and so on.

## Conclusion

China has become and will continue to be a country burdened with huge diabetes, which requires our government and society to pay more attention to the access to insulin, the life-saving medicine.

The results of this survey provide some base data on insulin availability and affordability in Nanjing area. We found the availability of insulin products was high in secondary hospitals and tertiary hospitals, but low in community hospitals. However, the affordability in community hospitals was better than other hospitals, but still unaffordable for those on low incomes.

The research on the price components of insulin products before and after the implementation of the policy showed the government has been working to regulate drug prices and availability. It is recommended that government improve access to insulin by increasing reimbursement rate, reducing prices and taxes, encouraging local manufacturers to develop and produce insulin products and provide appropriate subsidies to enhance their competitiveness with multinational companies.

## Supporting information

**S1 Table. Insulin price components before the implementation of the bidding policy.**
(XLSX)

**S2 Table. The proportion of insulin price components before the implementation of the bidding policy.**
(XLSX)

**S3 Table. Insulin price components after the implementation of the bidding policy.**
(XLSX)

**S4 Table. The proportion of insulin price components after the implementation of the bidding policy.**
(XLSX)

## Acknowledgments

The authors are very grateful to *Jiangsu Medicine Information Institute* for the data and support, and *Nanjing Pharmaceutical Co*., *Ltd f*or who willingly to provide their cooperation and support.

## Author Contributions

**Conceptualization:** Lulu Wang.

**Data curation:** Lulu Wang, Liang Dai.

**Formal analysis:** Liang Dai.

**Investigation:** Lulu Wang.

**Methodology:** Xin Li.

**Project administration:** Hui Liu, Weihong Ge.

**Resources:** Hui Liu, Huizhen Dai.

**Supervision:** Xin Li, Weihong Ge.

**Writing – original draft:** Lulu Wang.

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
