## [Decision Letter · Decision Letter 0]

23 Jul 2021

Availability, Affordability and Price Components of Insulin Products in Different-level Hospital Pharmacies: Evidence from Two Cross-Sectional Surveys in Nanjing, China

PONE-D-21-05946

Dear Dr. WANG,

We’re pleased to inform you that your manuscript has been judged scientifically suitable for publication and will be formally accepted for publication once it meets all outstanding technical requirements.

Kind regards,

Kevin Lu, PhD

Academic Editor

PLOS ONE

Reviewer's Responses to Questions

**Comments to the Author**

1. Is the manuscript technically sound, and do the data support the conclusions?

Reviewer #1: Yes

Reviewer #2: Yes

2. Has the statistical analysis been performed appropriately and rigorously? 

Reviewer #1: Yes

Reviewer #2: Yes

3. Have the authors made all data underlying the findings in their manuscript fully available?

Reviewer #1: Yes

Reviewer #2: Yes

4. Is the manuscript presented in an intelligible fashion and written in standard English?

Reviewer #1: Yes

Reviewer #2: Yes

5. Review Comments to the Author

Reviewer #1: All the data have been checked ,and the data were collected and analyzed correctly;

The whole paper were written surpported by the original data, and with qualified English level.

Condisering the significance of this paper and the contribution of this research,I recommend this paper.

Reviewer #2: The cross-sectional surveys covering a large number of pharmacies in China scientifically described the availability and affordability, driver of the affordability of insulin products in Nanjing, China. Stratification analysis by dosage, types of insulin and types of hospitals were conducted to show factors that may impact availability and affordability. The study could be limited by the survey location in Nanjing, and may not be generalizable to other parts of China, which the author already stated in the limitation section.

6. PLOS authors have the option to publish the peer review history of their article (what does this mean?). If published, this will include your full peer review and any attached files.

Reviewer #1: No

Reviewer #2: No

---

## [Editor Report · Acceptance letter]

3 Aug 2021

PONE-D-21-05946 

Availability, Affordability and Price Components of Insulin Products in Different-level Hospital Pharmacies: Evidence from Two Cross-Sectional Surveys in Nanjing, China 

Dear Dr. Wang:

I'm pleased to inform you that your manuscript has been deemed suitable for publication in PLOS ONE. Congratulations! Your manuscript is now with our production department. 

Kind regards, 

on behalf of

Professor Kevin Lu 

Academic Editor

PLOS ONE